# `calcQPI`: A versatile tool to simulate quasiparticle interference

P. Wahl[1,2]$\star$, L.C. Rhodes[1] and C.A. Marques[1]

**1** SUPA, School of Physics and Astronomy, University of St Andrews, North Haugh, St Andrews, KY16 9SS, United Kingdom
**2** Physikalisches Institut, Universität Bonn, Nussallee 12, 53115 Bonn, Germany

$\star$ wahl@st-andrews.ac.uk

## Abstract

Quasiparticle interference imaging (QPI) provides a route to characterize electronic structure from real space images acquired using scanning tunneling microscopy. It emerges due to scattering of electrons at defects in the material. The QPI patterns encode details of the $k$-space electronic structure and its spin and orbital texture. Recovering this information from a measurement of QPI is non-trivial, requiring modelling not only of the dominant scattering vectors, but also the overlap of the wave functions with the tip of the microscope. While, in principle, it is possible to model QPI from density functional theory (DFT) calculations, for many quantum materials it is more desirable to model the QPI from a tight-binding model, where inaccuracies of the DFT calculation can be corrected. Here, we introduce an efficient code to simulate quasiparticle interference from tight-binding models using the continuum Green's function method.

# 1  Introduction

Quasiparticle interference imaging (QPI) by scanning tunneling microscopy provides information about the low energy electronic structure of quantum materials with very high energy resolution, however its interpretation is usually highly non-trivial. Quasiparticle interference is due to scattering of itinerant electronic states close to the Fermi energy from defects, encoding information about the dominant scattering vectors **q** connecting, in the simplest approximation, parallel parts of the Fermi surface. The complexity arises because in reality, also the orbital and spin texture of the initial and final states play a role in determining the intensity of these patterns. Inverting these patterns to recover the electronic structure in **k** space is often non-trivial and not unique — therefore in principle angle resolved photoemission spectroscopy (ARPES) provides a much more direct measurement of the electronic structure. However, quasiparticle interference can provide valuable additional information either in parameter regimes not accessible by ARPES, or where the scattering matrix elements encode additional information. With regards to parameter regimes, ARPES is limited to the occupied states, so far to an energy resolution of typically about 1meV and measurements in zero magnetic field. Quasiparticle interference can probe materials with an energy resolution limited

practically only by temperature, and in magnetic fields limited only by what is technically feasible, i.e. at the moment up to about 30T [1], and probe occupied and unoccupied states in the same measurement. It can provide information about the detailed electronic structure close to the Fermi energy in the phase space relevant for many materials exhibiting quantum critical phenomena and quantum phase transitions [2,3], as well as for unconventional superconductors. Moreover, in the case of unconventional superconductors, the QPI patterns in the energy range of the superconducting gap encode information about the sign structure of the superconducting order parameter. Therefore, this Bogoliubov quasiparticle interference (BQPI) is one of a few techniques that can provide phase-sensitive information about the symmetry of a superconducting order parameter. However, much like for the interpretation of QPI patterns in the normal state of the materials, an interpretation of the BQPI patterns requires detailed modelling accounting for the surface and the overlap of the wave functions with the tip of the microscope.

Traditionally, the interpretation of QPI patterns was done based on lattice Green's function calculations, which simulate the change in the density of states due to the presence of a defect from a scattering approach and using a lattice Green's function. The technique therefore provides one value for the density of states per lattice point, neglecting the structure within the unit cell and the wave function overlap with the tip. This meant that at best, these calculations show qualitatively similar patterns compared to experimental data. The continuum Green's function technique pioneered by Choubey, Kreisel and Hirschfeld [4–8] provides an elegant combination of *ab-initio* modelling of the wave function overlap with the tip, and an ability to start from tight-binding models that allow for addition of different terms to the Hamiltonian beyond what is captured by density functional theory (DFT). Here, we introduce an implementation of the continuum Green's function method which can simulate scattering of quasiparticles at a defect from a given tight-binding model using the T-matrix approach to calculate differential conductance maps, but also can be used to calculate the spectral function in an extended zone scheme. The `calcQPI` code provides an efficient implementation of the continuum Green's function approach, fully parallelized using multi-threading via openMP and multi-processing via MPI, as well as GPU implementations for NVidia, AMD and Apple GPUs. It has been used in a number of recent works, demonstrating the robustness of the code [9–16] as well as its applicability in materials exhibiting magnetism and superconductivity, and for multi-band systems. This paper is meant to serve as a reference for the code, highlighting key features as well as providing key aspects of the implementation, to make it useful also for other researchers.

## 2 Theory

We model QPI using the continuum Green's function method [4,5,8]. In the following, we will describe the key components of the underpinning theory as far as it is required to understand the calculations which can be performed by `calcqpi` and the output.

### 2.1 Tight-binding model and wave functions

Key input for calculating the continuum Green's function is a tight-binding model and corresponding Wannier functions that connect momentum and continuous-real space. Both can be obtained from *ab-initio* calculations using DFT and then tools such as Wannier90 [17], or by other methods, including, e.g., from Slater-Koster tables [18] and approximating Wannier functions via spherical harmonic functions. The input for `calcQPI` expects the tight-binding model in the standardised Wannier90 format [17], but is otherwise agnostic about the origin of the model.

### 2.1.1 Tight-binding model

The standard way the tight-binding models are fed into a calculation is from a text file in Wannier90 output form (`wannier90_hr.dat`). Apart from a short header, the main content of the file are the hopping terms of a homogeneous system $\hat{t}_{\mathbf{R}}$ for lattice vectors $\mathbf{R} = x \cdot \mathbf{a}_1 + y \cdot \mathbf{a}_2 + z \cdot \mathbf{a}_3$ (where $\mathbf{a}_i$ are the unit cell vectors and $x$, $y$, and $z$ integer values), including the on-site terms (at $\mathbf{R} = (0, 0, 0)$). From these, the full Hamiltonian can be constructed from a sum over the nearest neighbour unit cells in real space

$$\hat{H}(\mathbf{k}) = \sum_{\mathbf{R}} \hat{t}_{\mathbf{R}} e^{i\mathbf{k}\cdot\mathbf{R}}, \tag{1}$$

and where $\hat{H}(\mathbf{k})$ and $\hat{t}_{\mathbf{R}}$ are matrices spanning the electronic degrees of freedom, i.e. orbital $\mu$ and spin $\sigma$, as well as, in the superconducting case, both particle and anti-particle terms. For magnetic models, `calcQPI` assumes that the hopping terms are sorted by spin component, that is, that the upper half of the terms is spin up and the lower half spin down. To indicate that the tight-binding model is magnetic, the keyword `spin=true;` needs to be included in the `calcQPI` input file (see table 2).

### 2.1.2 Superconducting models

Superconductivity is handled using the same type of input as normal tight-binding models, except that the input is interpreted as in the Nambu formalism, i.e. the input file encodes the particle and anti-particle sector of the Nambu spinors, and includes the superconducting order parameter in real space. This is then used to reconstruct the Bogoliubov-de Gennes Hamiltonian as

$$\hat{H}_s = \begin{bmatrix} \hat{H}(\mathbf{R}) & \hat{\Delta}(\mathbf{R}) \\ \hat{\Delta}^\dagger(\mathbf{R}) & -\hat{H}^T(-\mathbf{R}) \end{bmatrix}, \tag{2}$$

where again a homogeneous system is assumed. We note that treating the superconducting gap homogeneous here will neglect the influence of the defect on the local gap size which is accounted for, for example, in refs. [4, 5] by solving the inhomogeneous Bogoliubov-de-Gennes equations including the defect. Because the input file for the tight-binding model in Wannier90 format only contains the hopping terms, the code needs to be explicitly made aware if a model is superconducting, by including the keyword `scmodel=true;`. The convention used in `calcQPI` is that the first set of states are all the particle states, including spin and orbital degree of freedoms, and then all antiparticle states. The order parameter $\Delta$ is included in the off-diagonal terms connecting the particle and anti-particle sectors, and can include also complicated spatial form factors via pairing terms on the $\mathbf{R} \neq 0$ hopping terms (see, e.g., example 4.7).

### 2.1.3 Wave functions

For the wave functions for the spatial overlaps, either atomic-like wave functions can be used, or the output from the DFT calculation from which the tight-binding model has been projected via Wannier90 (see table 3).

For atomic-like wave functions, in `calcQPI`, Slater-type atomic orbitals for $s$, $p$, $d$ and $f$ orbitals are implemented to model the wave function overlap. They have a spatial decay of $\sim e^{-\alpha \mathbf{r}}$ which ensures that here the dependence of the tunneling current on the tip-sample distance $z$ recovers the correct physical behaviour, i.e. $I \propto e^{-\kappa z}$. The decay constant $\alpha$ is a free parameter that controls the radius of the orbital. The order of the orbitals provided to `calcQPI` in must be the same as is used in the tight-binding model.

To use wave functions projected from Wannier90, it is important to ensure that the phases between the wave functions remain correct - by default, Wannier90 normalizes the wave functions by the value which has the maximum modulus, which results in a randomization of the sign of the wave functions. This can be easily fixed by a patch of Wannier90 that retains the relative sign between the wave functions [19].

## 2.2 Obtaining the Green's function

We model the QPI starting from the unperturbed Green's function $\hat{G}_0$. There are multiple ways in which this can be obtained. The standard way in the QPI calculations here is from a two-dimensional tight-binding model, however there are other possibilities. In `calcQPI`, in addition, calculation of a surface and bulk Green's function from a 3D tight-binding model is supported to provide calculations in cases where either the inter-layer coupling can not be neglected, or to model surface states.

### 2.2.1 'Normal' Green's function

The standard treatment starts from a 2D tight-binding model. From the tight-binding model introduced in section 2.1.1, we calculate the momentum space lattice Green's function of the unperturbed host,

$$\hat{G}_{0,\sigma}(\mathbf{k}, \omega) = \sum_n \frac{\xi_{n\sigma}^\dagger(\mathbf{k})\xi_{n\sigma}(\mathbf{k})}{\omega - E_{n\sigma}(\mathbf{k}) + i\eta}, \tag{3}$$

where $\mathbf{k}, \omega$ define the momentum and energy, $\xi_{n\sigma}(\mathbf{k})$ and $E_{n\sigma}(\mathbf{k})$ are the eigenvectors and eigenvalues of the tight-binding model with band index $n$ and spin $\sigma$. The $\eta$ (specified by `eta`) adds a small imaginary part in the denominator which ensures the Green's function remains an analytical function when $\omega = E_{n\sigma}(\mathbf{k})$ and mimics a lifetime broadening. It can be used to emulate the resolution of the experiment (e.g. thermal and lock-in broadening). In the `calcQPI` input file, this mode is selected by including the keyword `green=normal;`, and it is the default mode when the keyword `green` is not included explicitly.

### 2.2.2 Surface and bulk Green's function

To simulate surface states from a surface Green's function, the `calcQPI` code uses the iterative surface Green's function approach introduced in refs. [20–22] and also used by the Wannier-Tools [23]. For this method to work, the tight-binding model must be brought into a principal layer form, where the out-of-plane hoppings are at most to the next adjacent layer, i.e. to $R_z = \pm 1$. In practice, this is achieved by building a super cell of a given tight-binding model along the $z$-direction until only the nearest-neighbour hoppings are left. We will here only provide a very brief sketch of the method, and for a full account refer the interested reader to the original literature [22].

The method uses a recursive scheme where an effective in-plane Hamiltonian $\hat{h}_N^{\text{s,b}}(\mathbf{k}_\parallel)$ for a slab of thickness $2^N$ for the surface (s) and the bulk (b) is calculated after $N$ iterations until convergence is reached and from which the surface and bulk Green's functions are obtained. The effective Hamiltonian is calculated by splitting the Hamiltonian of the principal layer $H_p$ into an in-plane part $\hat{H}_0^\parallel = \hat{H}_0(\mathbf{k}_\parallel, R_z = 0)$ and the out-of-plane parts $\hat{\alpha}_0 = \hat{H}_0^\perp(\mathbf{k}_\parallel) = \hat{H}_0(\mathbf{k}_\parallel, R_z = +1)$ and $\hat{\beta}_0 = \hat{H}_0^{\perp\dagger}(\mathbf{k}_\parallel) = \hat{H}_0(\mathbf{k}_\parallel, R_z = -1)$. The starting point is the in-plane Hamiltonian, i.e. $\hat{h}_0^{\text{b,s}}(\mathbf{k}_\parallel) = \hat{H}_0^\parallel(\mathbf{k}_\parallel)$. At each step, the out-of-plane couplings are adjusted to reflect the thicker layer through $\hat{\alpha}_N = \hat{\alpha}_{N-1}(\omega\mathbb{1} - \hat{h}_{N-1}^{\text{b}}(\mathbf{k}_\parallel))^{-1}\hat{\alpha}_{N-1}$ (and likewise for $\hat{\beta}_N$, note that $\omega$ here includes the small imaginary part $i\eta$, written explicitly in eq. 3). The effective Hamiltonians for

the bulk, $\hat{h}_N^{\mathrm{b}}(\mathbf{k}_\parallel)$, and for the surface, $\hat{h}_N^{\mathrm{s}}(\mathbf{k}_\parallel)$, when doubling the layer thickness between step $N-1$ and $N$, are built from

$$\hat{h}_N^{\mathrm{b}}(\mathbf{k}_\parallel) = \hat{h}_{N-1}^{\mathrm{b}}(\mathbf{k}_\parallel) + \hat{\alpha}_{N-1}(\omega\mathbb{1} - \hat{h}_{N-1}^{\mathrm{b}}(\mathbf{k}_\parallel))^{-1}\hat{\beta}_{N-1} + \hat{\beta}_{N-1}(\omega\mathbb{1} - \hat{h}_{N-1}^{\mathrm{b}}(\mathbf{k}_\parallel))^{-1}\hat{\alpha}_{N-1} \quad (4)$$

$$\hat{h}_N^{\mathrm{s}}(\mathbf{k}_\parallel) = \hat{h}_{N-1}^{\mathrm{s}}(\mathbf{k}_\parallel) + \hat{\alpha}_{N-1}(\omega\mathbb{1} - \hat{h}_{N-1}^{\mathrm{b}}(\mathbf{k}_\parallel))^{-1}\hat{\beta}_{N-1} \quad (5)$$

until convergence is reached, i.e. until $\hat{h}_N^{\mathrm{b,s}}(\mathbf{k}_\parallel) \sim \hat{h}_{N-1}^{\mathrm{b,s}}(\mathbf{k}_\parallel)$. In practice convergence is reached when the residual out-of-plane couplings between the effective layers, $\hat{\alpha}_N$ and $\hat{\beta}_N$, are smaller than a certain threshold (in `calcQPI`, until the $L^2$-norm is smaller than a threshold, i.e. $|\hat{\alpha}_N|_2 <$ `epserr` and $|\hat{\beta}_N|_2 <$ `epserr`, see table 2).

The surface and bulk Green's functions $\hat{G}_{\mathrm{s}}(\mathbf{k}_\parallel, \omega)$ and $\hat{G}_{\mathrm{b}}(\mathbf{k}_\parallel, \omega)$ are obtained from

$$\hat{G}^{\mathrm{b,s}}(\mathbf{k}_\parallel, \omega) = (\omega\mathbb{1} - \hat{h}_N^{\mathrm{b,s}})^{-1}. \quad (6)$$

The surface Green's function describes the electronic states at the surface of the material, and the bulk Green's function those in the bulk in a plane parallel to the surface. Both can then be used to perform spectral function or QPI calculations [10]. We note that while this method is very useful for modelling surface states, the choice of wave functions for the continuum transformation is not straightforward.

To select the surface or bulk Green's function method for the calculation, one includes the keyword `green=surface` or `green=bulk`, respectively, in the `calcQPI` input file (see table 2). The selected Green's function will be used in subsequent QPI calculations or calculations of the spectral function. The implementation is built on top of the Hamiltonian as defined in section 2, which means that the surface Green's function calculation can also be done for superconducting models.

## 2.3 $T$-matrix calculation

We Fourier transform $\hat{G}_{0,\sigma}(\mathbf{k}, \omega)$ to obtain the unperturbed real space lattice Green's function, $\hat{G}_{0,\sigma}(\mathbf{R}, \omega)$, and follow the usual $T$-matrix formalism to obtain the Green's function of the system including an impurity from

$$\hat{G}_\sigma(\mathbf{R}, \mathbf{R}', \omega) = \hat{G}_{0,\sigma}(\mathbf{R} - \mathbf{R}', \omega) + \hat{G}_{0,\sigma}(\mathbf{R}, \omega)\hat{T}_\sigma(\omega)\hat{G}_{0,\sigma}(-\mathbf{R}', \omega), \quad (7)$$

where the $\hat{T}_\sigma$-matrix, given by

$$\hat{T}_\sigma = \hat{V}_\sigma \left(\mathbb{1} - \hat{V}_\sigma \hat{G}_{0,\sigma}(0, \omega)\right)^{-1}, \quad (8)$$

describes the scattering at the impurity. It is typically sufficient to consider a point-like non-magnetic impurity with equal scattering strength in the spin-up and spin-down channels, such that $\hat{V}_\sigma = V_0\mathbb{1}$. `calcQPI` does allow the scattering strength of different orbital or spin channels to be set independently. To set the properties of the scattering potential, the keyword `scattering` and `phase` need to be specified in the `calcQPI` input file (see table 2). The impurity potential can also be obtained from *ab-initio* calculations [6], however it often has a minor influence on the appearance of the QPI patterns compared to wave function overlaps.

The Green's function calculation will be performed on a discrete real-space lattice made of `lattice` number of unit cells, as in the example shown in fig. 1(a) for `lattice`= 32. In `calcQPI`, the impurity is located in the unit cell at $\mathbf{R} = (0,0)$, shown as grey square in fig. 1(a).

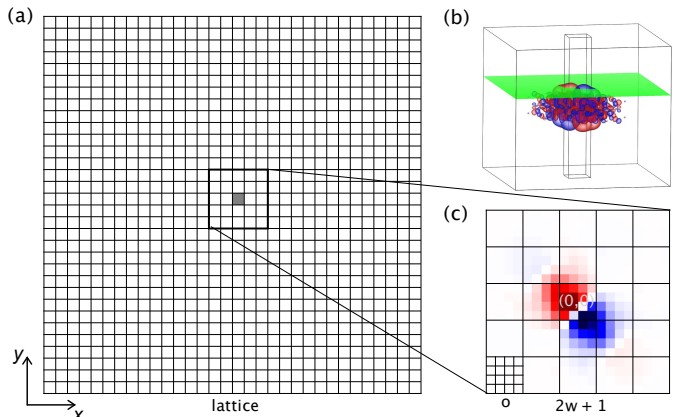

Figure 1: **Grids and wave functions.** (a) Main real-space lattice on which the QPI calculation is performed, shown here for `lattice`= 32. The defect is in within the grey area in the central unit cell, though the exact location depends on the model. The square indicated by a thick black line indicates the area covered by the `window` parameter $w$, here shown for $w = 2$. (b) Wannier function for a $d_{xz}$-orbital obtained from Wannier90. The green plane is at height `zheight`= 3 above the topmost atom. (c) Wave function file and discretization scheme. The wave function shows the cut indicated in (b) for the same wave function, which is used in the QPI calculation. The lattice is for `window` parameter $w = 2$ which sets an effective cut-off radius. On the left bottom unit cell the oversampling parameter `oversamp` is indicated - the lattice of unit cells is oversampled by $o$ values, here shown for $o = 4$.

## 2.4 Continuum transformation

To realistically model the QPI such that we can compare the results from calculations with experiment, we use the continuum Green's function approach [4, 5, 8], which defines the Green's function in terms of the continuous spatial variable $\mathbf{r}$ as

$$G_\sigma(\mathbf{r}, \mathbf{r}', \omega) = \sum_{\mathbf{R}, \mathbf{R}', \mu, \nu} G_\sigma^{\mu, \nu}(\mathbf{R}, \mathbf{R}', \omega) w_{\mathbf{R}, \mu}(\mathbf{r}) w_{\mathbf{R}', \nu}(\mathbf{r}'), \tag{9}$$

where $w_{\mathbf{R}, \nu}(\mathbf{r})$ are the Wannier functions connecting the local and lattice space. In the QPI calculation, only a section through the wave functions at a fixed height `zheight` above the surface is required, as illustrated by the green plane in fig. 1(b). The code needs to be told explicitly if the model is magnetic to ensure that the spin indices are excluded from the sum in eq. 9.

### 2.4.1 Cut-off radius

One of the computationally most expensive steps in the calculation is the continuum transformation in eq. 9, which consists of six nested loops over the in-plane lattice vectors $\mathbf{R}$ and $\mathbf{R}'$ and the orbital indices $\mu$ and $\nu$. However, it can be easily seen that only a subset of terms are important: the contribution of individual terms in the sums over $\mathbf{R}$ in eq. 9 will become negligible for large $|\mathbf{R} - \mathbf{R}'|$ because then the term $|w_{\mathbf{R}, \mu}(\mathbf{r}) w_{\mathbf{R}', \nu}(\mathbf{r}')|$ becomes very small due to the localized nature of the wave functions $w_{\mathbf{R}, \mu}(\mathbf{r})$. Therefore, effectively, the sum can be cut off to only contain terms $|\mathbf{R} - \mathbf{R}'| < r_{\text{cutoff}}$.

In `calcQPI`, there are two ways in which the efficient calculation of the sum is achieved. For the first one, which is particularly useful when the spatial extent of the wave functions

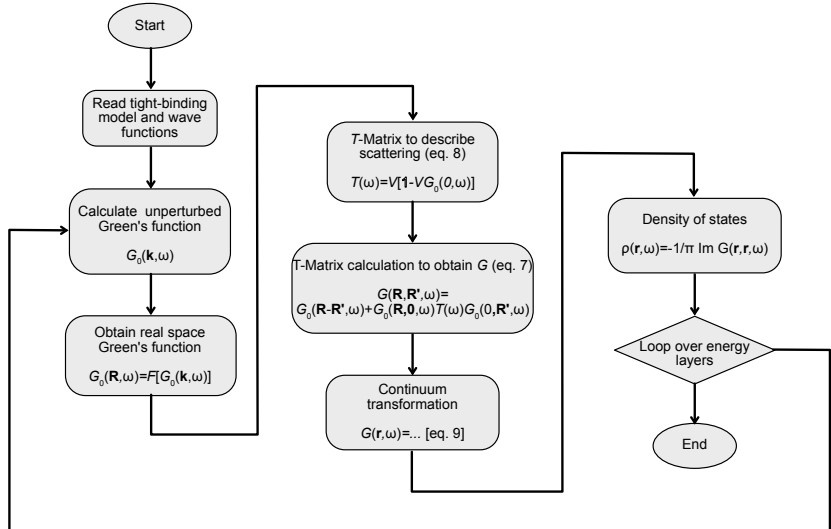

Figure 2: **Flowchart of** `calcQPI`. Flowchart summarizing the implementation of the continuum QPI calculation in calcQPI, starting from the real space Hamiltonian, via calculation of the unperturbed Green's function $\hat{G}_0(\mathbf{k}, \omega)$, the $T$-matrix calculation to describe the scattering and the continuum transformation to obtain $\hat{G}(\mathbf{r}, \mathbf{r}, \omega)$. The calculation is enclosed in a loop over the energy layers $\omega_n$ of the QPI map $\rho(\mathbf{r}, \omega)$.

$w_{\mathbf{R},\mu}(\mathbf{r})$ is larger than the size of the unit cell, the sums over $\mathbf{R}$ and $\mathbf{R}'$ are limited to only $w$ neighbouring unit cells (where $w$ is set by the keyword `window`) effectively introducing a cut-off radius. This is in practice achieved by running the summations over $\mathbf{R}$ and $\mathbf{R}'$ over a window $w$,

$$\mathbf{R}_{i,j} = \mathbf{R}_0 + \begin{pmatrix} i \\ j \end{pmatrix}, \quad \text{for } i, j = -w, \ldots, 0, \ldots, w, \tag{10}$$

thus covering $(2w+1) \times (2w+1)$ unit cells, as illustrated in fig. 1(c). In the example shown in fig. 1(c), one can see that the wave function shows negligible weight for the next-nearest neighbour unit cells, thus it is unnecessary to include further unit cells in the sum. This is efficient and a good approximation for systems with only a few atoms per unit cell.

For systems with large unit cells, such as moiré systems [15], where the wave functions $w_{\mathbf{R},\mu}(\mathbf{r})$ decay on a shorter length scale than the size of the unit cell, this cut-off scheme becomes inefficient as the sum in eq. 9 contains many terms with negligible contributions, e.g. for atoms at opposite ends of the moiré unit cell. This is circumvented by precomputing the products $w_{\mathbf{R},\mu}(\mathbf{r})w_{\mathbf{R}',\nu}(\mathbf{r})$ on the lattice and retaining only terms that are larger than a threshold $t$. In the implementation in `calcQPI`, this is achieved by creating a list with all non-negligible terms larger than a certain threshold $t$, i.e. $|w_{\mathbf{R},\mu}(\mathbf{r})w_{\mathbf{R}',\nu}(\mathbf{r})| > t$. This results in a dramatic speed-up for calculations for systems with large unit cells.

## 2.5 Continuum quasiparticle interference

From the continuum Green's function, eq. 9, the quasiparticle interference is obtained straightforwardly by calculating the density of states $\rho(\mathbf{r}, \omega)$ in real space from

$$\rho(\mathbf{r}, \omega) = -\frac{1}{\pi} \text{Tr}_\sigma \left( \text{Im} \hat{G}(\mathbf{r}, \mathbf{r}, \omega) \right), \tag{11}$$

where the trace is over the spin degrees of freedom. For the calculation of a QPI map, $\rho(\mathbf{r}, \omega)$ is calculated on a grid of $n \times n$ unit cells and $o \times o$ points per unit cell, where $o$ is defined as the oversampling (`oversamp`) within the unit cell, fig. 1(c). The number of unit cells $n$ is set by `lattice`. Thus, the image will have in total $(o \cdot n) \times (o \cdot n)$ pixels. Calculations are typically done for multiple energies in discrete layers, so that the result is a 3D data array. The resulting continuum local density of states is expected to be representative of the measured differential conductance at a fixed tip height, i.e. $g(\mathbf{r}, V) \sim \rho(\mathbf{r}, eV)$. The resulting $\rho(\mathbf{r}, \omega)$ map is then Fourier transformed to simulate the QPI map, $\tilde{\rho}(\mathbf{q}, \omega)$. Where the unit cell contains multiple atoms, it is advisable to consider the effect of scatterers at the different sites.

Fig. 2 shows the basic structure of how the continuum Green's function calculation is implemented in `calcQPI` based on eqs 1–11. To output this calculation, the keyword `output=wannier` is included in the input file.

### 2.5.1 Setpoint effect

While it is in principle possible to acquire spectroscopic maps at fixed tip-sample distance, from an experimental point of view, it is often more practical to readjust the tip-sample distance at each point of a map by briefly closing the feedback loop in between spectra to restabilize the $z$-height for a fixed voltage $V_0$ and current $I_0$ to then acquire the subsequent spectrum. This results in a setpoint effect, which effectively means that while the tunneling spectrum $g(\mathbf{r}, V)$ is proportional to the density of states $\rho(\mathbf{r}, eV)$, i.e. $g(\mathbf{r}, V) = \alpha(\mathbf{r})\rho(\mathbf{r}, eV)$, the proportionality constant $\alpha(\mathbf{r})$ is in general position-dependent [24]. This is not an issue when comparing features in individual spectra, however does introduce artifacts in the Fourier transformation and hence in the QPI. While the calculations as implemented in `calcQPI` do not account for this effect, it can be emulated to a very good approximation by normalizing $\rho(\mathbf{r}, eV)$ by the integral of the continuous density of states from the Fermi energy to a given setpoint voltage at each point. With $I_0(\mathbf{r}) = \int_0^{eV_0} \rho(\mathbf{r}, E)\mathrm{d}E$, one obtains

$$g_s(\mathbf{r}, V) = \frac{\rho(\mathbf{r}, eV)}{I_0(\mathbf{r})}. \tag{12}$$

In principle, the setpoint effect can also be properly modelled within the continuum Green's function approach by using a spatially varying $z$-height that is calculated at each point, as has been done, for example, in ref. [8].

### 2.6 Spectral function

To be able to fit a tight-binding model simultaneously to STM and ARPES data, it is useful to be able to also calculate the spectral function. In `calcQPI`, two different ways to calculate the spectral function are implemented, using either the lattice Green's function ('normal' spectral function) as well as, by using the positions of the atoms, an unfolded spectral function.

### 2.6.1 Normal spectral function

The spectral function is defined through

$$A(\mathbf{k}, \omega) = -\frac{1}{\pi}\mathrm{Tr}\left(\mathrm{Im}\hat{G}_0(\mathbf{k}, \omega)\right), \tag{13}$$

The spectral function is calculated by using the keyword `output=spf` (see table 1). We note that the output will depend on the choice of unit cell of the tight-binding model.

### 2.6.2 Unfolded spectral function

Where the tight-binding model contains several atoms within the unit cell, for example due to a moiré periodicity or a charge or spin density wave, this results in band foldings into a smaller Brillouin zone, rendering an interpretation of the band structure difficult when only considering **k**-states within the first Brillouin zone as would be the case in the 'normal' spectral function. Considering the electronic states in an extended zone scheme often facilitates a more direct comparison with ARPES and also QPI. To facilitate comparison of the electronic structure with STM and ARPES, `calcQPI` can also be used to calculate an unfolded spectral function. To obtain the unfolded spectral function from a tight-binding model, following ref. [15], we calculate

$$A(\mathbf{k}, \omega) = -\frac{1}{\pi} \sum_{\mu, \nu} e^{i\mathbf{k}(\mathbf{r}_\mu - \mathbf{r}_\nu)} \cdot \mathrm{Im} G_0^{\mu\nu}(\mathbf{k}, \omega), \tag{14}$$

where $\mu$, $\nu$ are orbital indices, and $\mathbf{r}_\mu$ are the intra-unit-cell positions of orbital $\mu$. These positions need to be specified separately in the input file, as they are not contained in the Wannier90-style file with the tight-binding model. To perform such a calculation, the `calcQPI` input file needs to include the keyword `output=uspf`.

## 2.7 Josephson tunneling

For superconducting Hamiltonians, the calculation of the Josephson critical current is implemented, accounting for matrix element effects. The calculation of the critical current $I_c(\mathbf{r})$ follows refs. [25–27]. From the Bogoliubov-de-Gennes equation, eq. 2, the full Green's function of the superconducting system is obtained, which can be expressed in terms of the normal and anomous Green's functions $\hat{G}(\mathbf{k}, \omega)$ and $\hat{F}(\mathbf{k}, \omega)$ (see, e.g., [28])

$$\hat{G}_s(\mathbf{k}, \omega) = \begin{bmatrix} \hat{G}(\mathbf{k}, \omega) & \hat{F}(\mathbf{k}, \omega) \\ -\hat{F}^\star(-\mathbf{k}, -\omega) & -\hat{G}^\star(-\mathbf{k}, -\omega) \end{bmatrix}. \tag{15}$$

$\hat{F}(\mathbf{r}, \mathbf{r}, \omega)$ is directly obtained from the continuum transformation in eq. 15. The critical current can be then obtained from

$$I_c = \int \frac{\mathrm{d}\omega}{2\pi} n_\mathrm{F}(\omega) \mathrm{Im}(\hat{F}^\dagger(\mathbf{r}, \mathbf{r}, \omega) \hat{F}_t(\omega)), \tag{16}$$

where $n_\mathrm{F}$ is the Fermi function and $\hat{F}_t^\dagger(\omega)$ the anomalous Greens function of the tip and here set to be

$$F_t(\omega) = -i\mathrm{sgn}(\omega) \frac{\pi N_0 \Delta_t}{\sqrt{(\omega + i\eta_t)^2 - \Delta_t^2}}, \tag{17}$$

assuming that the tip is an $s$-wave superconductor (with spin indices included this becomes $\hat{F}_t(\omega) = F_t(\omega) i\hat{\sigma}_y$). For the calculation, the gap size of the tip $\Delta_t$ and the broadening of the tip, $\eta_t$ are required. `calcQPI` calculates

$$p(\mathbf{r}, \omega) = \frac{1}{2\pi} \mathrm{Tr}_\sigma \mathrm{Im}(\hat{F}(\mathbf{r}, \mathbf{r}, \omega) \hat{F}_t(\omega)), \tag{18}$$

which then still needs to be integrated to obtain $I_c$, i.e. $I_c \sim \int_{\omega_0}^0 p(\mathbf{r}, \omega) \mathrm{d}\omega$. In practice, one will calculate a map $p(\mathbf{r}, \omega)$ in a range $-\omega_0 \ldots 0$, chosing an $\omega_0$ that is sufficiently large so that $p(\mathbf{r}, \omega_0) \ll I_c$. To select this mode, the keyword `output=josephson` is included in the output file, and the parameters detailed in table 5 for the tip need to be specified.

## 3  Technical implementation

`calcQPI` is written in C++, using the Gnu Scientific Library for linear algebra operations such as matrix operations, matrix inversions and to solve eigenvalue problems, and FFTW3 for Fast Fourier transformations. For parallelization, it uses openMP for multi-threading and MPI libraries to allow running multiple tasks on a node or across multiple nodes. A release version of the code is available at ref. [29].

### 3.1  CPU code

The CPU code is hybrid-parallelized using MPI for parallelization over tasks, and OpenMP for parallelization over threads. The main loop over energy layers is split over tasks, whereas the loops over the $k$-lattice and the real space lattice are parallelized using OpenMP. Therefore, for optimal performance, it is advisable that the number of energy layers is an integer multiple of the number of MPI tasks, and the number of lateral $k$ and lattice points each an integer multiple of the number of cores. Because OpenMP threads all perform calculations on the same layer, an increased number of cores does not require additional memory, whereas each task requires the same amount of memory, therefore it is in general preferable to run one or two tasks per node when running across multiple nodes.

### 3.2  GPU code

In addition to the CPU parallelization for the calculation of energy layers and to obtain the eigenvectors and eigenvalues, the calculation of the continuum Green's function and spectral function (eqs. 11 and 14) can be performed on a GPU. This can result in significant speed ups, as the calculation can be done in parallel with calculating the Green's function of the next energy layer on the CPU. The GPU version is implemented for Nvidia, AMD and Apple GPUs, each effectively using the same code base. The Apple/Metal version runs only using single precision floats for the calculation, whereas the other GPU versions use double precision floats. Use of multiple GPUs is supported, the code assumes that there is one GPU per task.

We note that the GPU version will only result in efficient use of both GPU and CPU where there is a good load balancing between the parts calculated on each of them. This is typically the case for systems where the eigenvectors and eigenvalues are precomputed, however not for calculations using the surface or bulk Green's function or for calculations with large unit cells such as moiré systems. If one wants to ensure efficient use, this needs to be tested in individual cases.

### 3.3  Timings

Fig. 3 shows the quality of the parallelization for a choice of ideal parameters for the number of $k$-points, layers and the size of the lattice. The calculations shown here were performed using a model with 12 orbitals in total, including spin, for a $t_{2g}$ manifold and with two atoms in the unit cell (as is, for example, applicable for the surface layer of $Sr_2RuO_4$ [30]). The parameters used for the calculations were a lattice with 128 unit cells, $2048 \times 2048$ $k$-points, an oversampling of $o = 4$ and a window of $w = 2$. Calculations shown in Fig. 3 have been done with 128 (a), 144 (b) and 256 (c) energy layers. Calculations on GPUs have been done on nodes with four NVidia Tesla V100-SXM2-16GB GPUs each. The GPU nodes have two 20-core Intel Xeon Gold 6148 (Cascade Lake) CPUs with 2.5GHz and 384GB RAM, for each requested GPU 10 CPU cores are allocated. The comparison for a CPU node is for a node with two 2.1GHz Intel Xeon E5-2695 (Broadwell) processors with 18 cores each and 256GB of RAM. For the calculation with 16 GPUs, the overhead for data transfers starts to become a

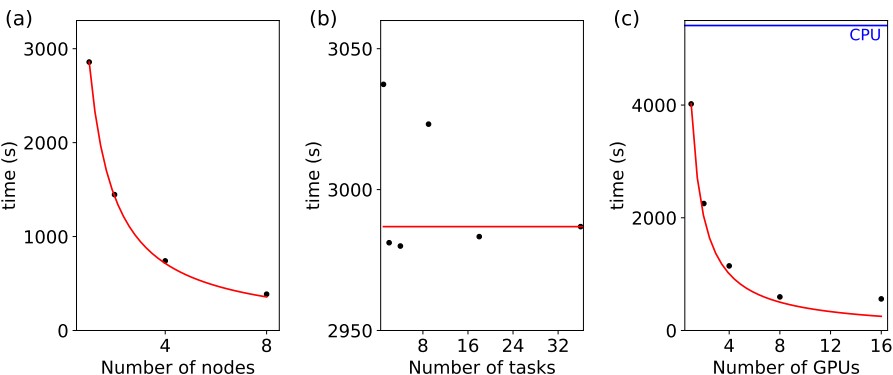

Figure 3: **Parallelization.** (a) Run time using 1, 2, 4 or 8 nodes and two tasks per node, showing near-perfect scaling and demonstrating quality of the MPI parallelization. (b) Benchmarking of the hybrid parallelization. The graph shows the time for a given QPI calculation when using one node with 36 cores, but for different number of MPI tasks $n$. Each task is given $36/n$ cores. There is only very little deviation from ideal behaviour, which would be a constant line. (c) Run time for the GPU version, using 1, 2, 4, 8 or 16 GPUs on up to four nodes (and 10 cores per task/GPU). Blue solid line shows the time for the same calculation running on one CPU node with the same number of layers as for the calculations done on a GPU. All calculations were done on the tier-2 supercomputer Cirrus.

significant part of the total time for the calculation, showing less than ideal scaling. In a direct comparison, the calculation on one GPU is about 25% faster compared to a CPU node, while for a full GPU node, the calculation is four times faster. The speed up for calculations on the GPU node depends on details of the calculation, so needs to be tested for individual cases.

## 4 Examples

Beyond published work [9–15], where `calcQPI` has been applied to realistic systems and compared to experimental QPI data where available, here we show a range of minimal models which show the capabilities of the code but can also serve to understand complex phenomena from tight-binding models that contain only the most important terms. As the last example we show calculations for a realistic multi-orbital system using a DFT-derived tight-binding model and wave functions and compare them to experimental QPI data. The minimal examples are available at ref. [31].

### 4.1 Nearest-neighbour tight-binding model

As a simple starting point, we use a nearest-neighbour tight-binding model (fig. 4(a)) on a square lattice in 2D. We only consider one hopping parameter $t$, which is set to $t = 0.1$eV, and leave the on-site energy $\varepsilon_0 = 0$eV. The spectral function at the Fermi surface is shown in fig. 4(b), showing only the square shaped Fermi surface of the model at half-filling. The band structure along $X - \Gamma - M$ is shown in fig. 4(c). To calculate the continuum local density of states, we model the wave function through a single $s$ orbital with radius 0.5, and for the tip height use 0.5. The radius is in units of the lateral size of the unit cell, whereas the height is relative to the radius (i.e. enters as $h/r$ in the wave function). With these parameters, a window $w = 2$ and oversampling $o = 4$ are sufficient to get a sufficiently high sampling of

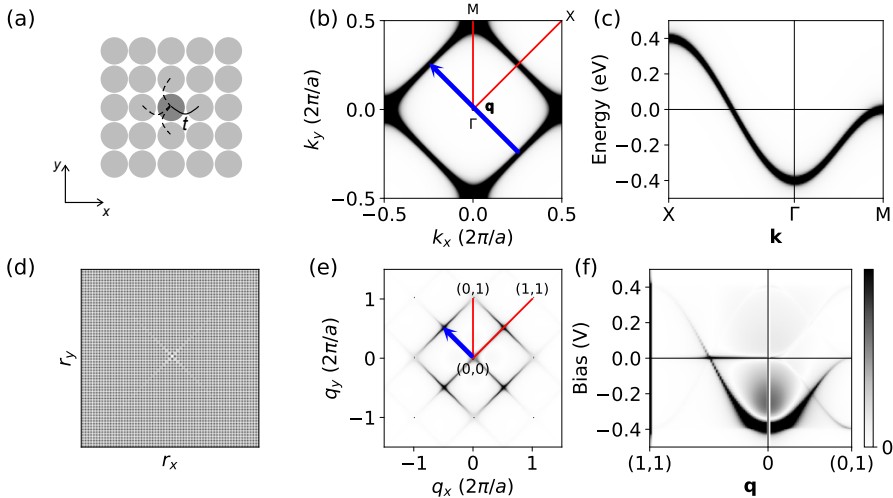

Figure 4: **QPI of a nearest-neighbour tight-binding model.** (a) Minimal tight-binding model, containing only an on-site energy $\epsilon_0 = 0$eV and nearest neighbour hopping term $t = -0.1$eV. (b) Constant energy cut of the spectral function at the Fermi energy $E = 0$eV, a blue arrow indicates a dominant nesting vector $\mathbf{q}$, and (c) cut along $X - \Gamma - M$ (see (b) for definition of symbols). (d) Real space image of the continuum density of states $\rho(\mathbf{r}, E)$ with a defect at the centre. (e) Fourier transform $\tilde{\rho}(\mathbf{q}, E)$ of (d), the dominant scattering vector $\mathbf{q}$ (blue arrow) is the same connecting parallel sheets of the Fermi surface in (b). (f) Energy cut through the simulated spectroscopic map.

the unit cell, and avoid artifacts due to the cut-off. In fig. 4(d), we show the resulting real space image of the continuum density of states as it would be measured in an STM in constant height mode. The atomic lattice and the defect can be seen, as well as signatures of the quasiparticle interference. These become clearer in the Fourier transformation, fig. 4(e), where the patterns corresponding to the characteristic scattering vectors are observed, recovering the square shape of the Fermi surface (see also blue arrows indicating a vector that connects parallel sections of the Fermi surface and dominates the QPI). Energy cuts through the QPI map, fig. 4(f), show evidence of the matrix element effect which results in reduced sensitivity for larger wave vectors $q$.

## 4.2 Nearest-neighbour tight-binding model on a hexagonal lattice

The `calcQPI` code can also straightforwardly be used with non-square lattices. We show in fig. 5 a QPI calculation for a nearest neighbour tight-binding model on a hexagonal lattice, with hopping term $t = -0.1$eV. Instead of performing the $k$-point sum over the hexagonal unit cell, the calculation is done over the area of a rhombus, fig. 5(b), which is equivalent. The resulting images of $\rho(\mathbf{r}, E)$ are skewed, and to recover the hexagonal shape of the lattice, they need to be unskewed according to the direction of the unit cell vectors, resulting in the real space continuous density of states of a hexagonal lattice, fig. 5(c). The Fourier transform $\tilde{g}(E, \mathbf{q})$ shows the atomic peaks corresponding to the atomic lattice, and the hexagonal shaped scattering vectors.

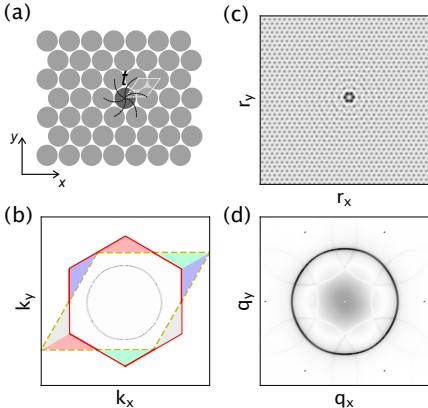

Figure 5: **QPI of a nearest-neighbour tight-binding model on a hexagonal lattice.**
(a) Minimal tight-binding model, now with three nearest neighbour hopping terms
$t = -0.1$eV. (b) Constant energy cut through the spectral function at the Fermi
energy $E = 0$eV. The red hexagon shows the conventional Brioullin zone, and the
dashed yellow rhombus the area for the $k$-point sum used in `calcQPI`, which can be
seen to be identical. Coloured areas show identical parts of $k$-space. (c) Real space
simulated continuum QPI $\rho(\mathbf{r}, E)$ at the Fermi energy with a defect in the centre. (d)
Fourier transform $\tilde{\rho}(\mathbf{q}, E)$ of (d).

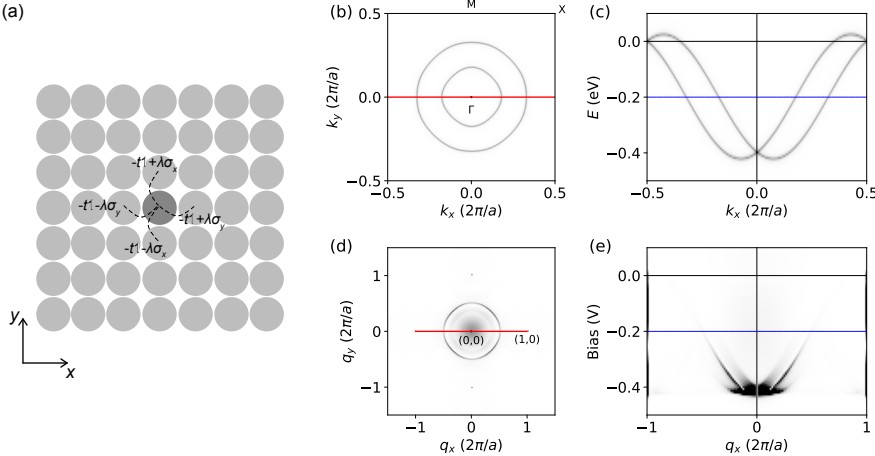

Figure 6: **QPI for a Rashba system.** (a) minimal model to capture Rashba physics in
a tight-binding model, including spin-flip terms on the nearest-neighbour hoppings.
(b) Spectral function of a Rashba system, assuming a nearest-neighbour tight-binding
model with $t = -0.1$eV and including a spin-flip term $\lambda\sigma_{y/x}$ for hopping along
x/y to model Rashba spin-orbit coupling. The spectral function is shown here for
$E = -0.2$eV. The two Rashba bands can be clearly seen. (c) Corresponding energy
cut along the red line in (b), showing the dispersion relation. The energy of the
constant energy cut shown in (b) is indicated by a blue dashed line. (d, e) Simulated
quasiparticle interference, showing a cut (c) at the same energy as in (b) and the QPI
dispersion (d) along the red line in (d).

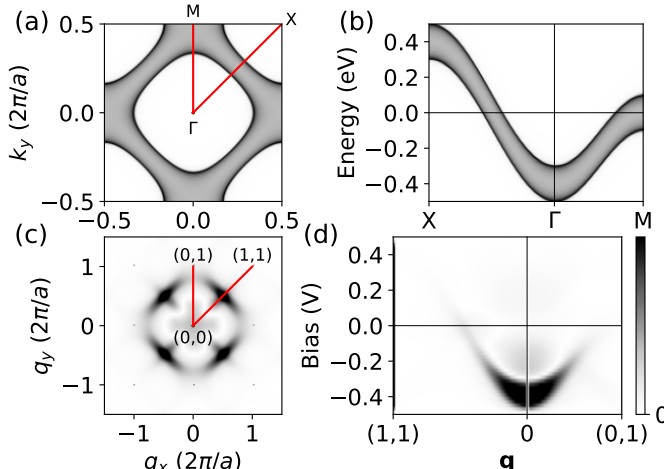

Figure 7: **QPI of a nearest-neighbour tight-binding model with out-of-plane hopping.** (a) Spectral function at the Fermi energy $E = 0$eV for model as in section 4.1, including an out-of-plane hopping $t_z = \frac{1}{2}t$. The spectral function has been calculated using the bulk Green's function $\hat{G}_{\rm b}(\mathbf{k}, \omega)$. (b) Cut of the spectral function along $X - \Gamma - M$ (see (b) for definition of symbols). (c) Fourier transform $\tilde{\rho}(\mathbf{q}, E)$ of the continuum density of states calculated from the surface Green's function $\hat{G}_{\rm s}(\mathbf{k}, \omega)$, (d) Energy cut through the simulated map along path indicated in c.

## 4.3 Rashba effect

Many surfaces exhibit surface states that are spin-split due to spin-orbit coupling and the Rashba effect. The potential application in spintronics applications has led to intense interest in characterising these states. To model such a state with spin splitting, we start from a Hamiltonian as in section 4.1, however now explicity including spin. The Rashba spin splitting of the form $\lambda(\hat{\sigma}_x k_y + \hat{\sigma}_y k_x)$ is accounted for by adding a spin flip term to the nearest neighbour hopping term, so that it becomes $t \cdot \mathbb{1} + \lambda \sigma_y$ for hopping in the $+x$ direction, where $\sigma_y$ is the Pauli matrix (and likewise $t \cdot \mathbb{1} + \lambda \sigma_x$ for hopping in the $+y$-direction). In QPI measurements of Rashba spin-split surface states, the Rashba spin splitting can usually not be resolved, because interference is due to non-orthogonality of the wave functions, which means that the lifting of spin degeneracy due to spin-splittings is not resolved [32]. This is also what we find here. In fig. 6 we show a QPI calculation for a minimal model of Rashba spin splitting, showing the suppressed scattering between bands with opposite spin.

## 4.4 Effects of $k_z$ dispersion on QPI

To demonstrate the effect of a $k_z$ dispersion on QPI, we introduce an out-of-plane hopping term $t_z = t/2$ into the tight-binding model [10]. The resulting spectral function obtained from the bulk Green's function $\hat{G}_{\rm b}(\mathbf{k}, \omega)$ is shown in fig. 7(a). It closely resembles the one of the purely two-dimensional model (fig. 4(b)), except that the constant energy contour now has contributions from a range of $k_z$ values. Similarly, the dispersion relation in the spectral function is now a broad ribbon (fig. 7(b)). This broadening is also reflected in the QPI obtained from the surface Green's function, $\hat{G}_{\rm s}(\mathbf{k}, \omega)$, fig. 7(c,d)), where the signal is substantially broadened. We note that the out-of-plane hopping here is a factor of two smaller than the in-plane hopping, which means that for more isotropic materials this $k_z$-'smearing' will be significantly larger. Such broadening is also seen in real materials [10, 33, 34].

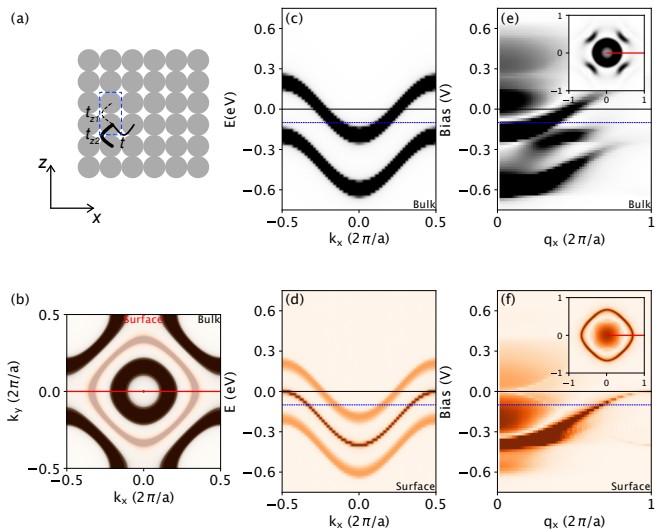

Figure 8: **QPI of a surface state.** (a) simple SSH-based model for a surface state - along the $z$-direction, the material is modelled as an SSH chain with alternating hoppings $t_{z1} = \frac{1}{2}t$ and $t_{z2} = 2t$, while the $x$- and $y$-directions are based on a simple nearest neighbour tight-binding model. Blue box indicates the unit cell. (b) Bulk (left half) and surface (right half) spectral function of the model in (a) at $E = -0.1\text{eV}$ showing two bulk bands, and the surface state between them in red. The surface state exists only in the surface spectral function. (c) Cut of the bulk spectral function along the red line in (b), (d) same cut through the surface spectral function, showing the dominant contribution of the surface state. (e, f) QPI of the bulk (e) and surface (f), showing the clear signature of the surface state in (f).

## 4.5 QPI of surface states

The surface Green's function approach discussed in section 2.2.2 allows for calculations of the QPI of surface states, such as, for example, the quasi2D surface states that exist at Cu(111), Ag(111) and Au(111) surfaces. For simplicity, we here introduce a minimal model to demonstrate such a calculation. For this, we write a simple tight-binding model with two orbitals per unit cell that mimics the Su–Schrieffer–Heeger (SSH) model in the $z$-direction by alternating the out-of-plane coupling between $t_{z1}$ and $t_{z2}$, and keeping the nearest neighbour model in the $x$-/$y$-directions as $t$, fig. 8(a). The SSH model exhibits a surface state when terminated between the stronger bound pair of atoms, here given by $t_{z2}$. The resulting surface state and its quasiparticle interference are shown in fig. 8: The bulk spectral function shows the two bulk bands that exist due to the two in-equivalent out-of-plane hoppings fig. 8(b, c). The surface spectral function shows a sharp surface state inbetween, fig. 8(b, d). While the QPI for the bulk Green's function is relative broad with no sharp features (fig. 8(e)), the QPI for the surface Green's function shows the surface state as a sharp features dominating the surface QPI (fig. 8(f)).

## 4.6 Topological surface states

The strength of the surface Green's function approach is that it also enables QPI calculations for topological surface states with protected back-scattering. Here, we introduce a minimal model based on the Bernevig-Hughes-Zhang model [35] that exhibits such a topological surface state, Fig. 9(a). Fig. 9(b, c) shows the resulting calculations of the spectral function of the bulk and

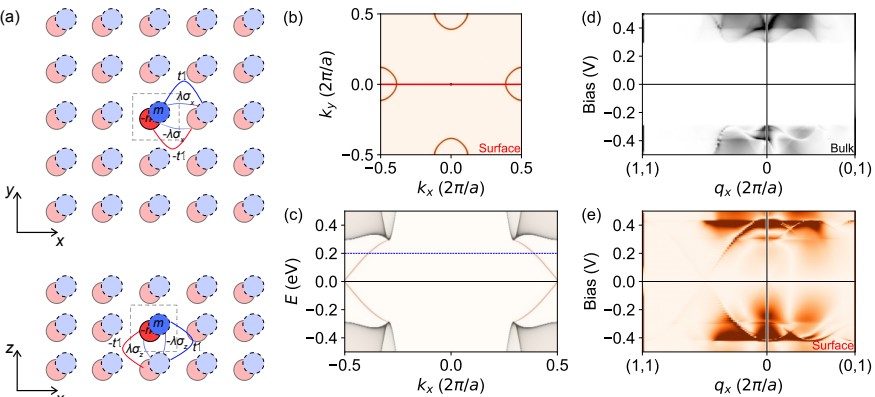

Figure 9: **QPI of a topological surface state.** (a) Schematic of minimal tight-binding model for a topological insulator. The blue and red shaded circles indicate two different orbitals on the same site. The upper half shows the spin-flip hopping terms along the x-direction, and the lower half along the z-direction. (b) Surface spectral function of the minimal model for a topological insulator. Only the surface state can be seen at the M points in the Brillouin zone. The energy is in the topological gap so there would be no contribution from the bulk states. (c) Cut through the Brillouin zone along the red line in (b), showing the bulk band gap and the topological surface state. Bulk (grey) and surface (red) spectral functions are shown superimposed. (d) QPI calculation based on the bulk Green's function and (e) based on the surface Green's function, showing that the topological surface state only results in a rather broad background around $\Gamma$, because direct backscattering is forbidden. In the $(1, 1)$ direction, some scattering from the topological state appears due to interpocket scattering vectors.

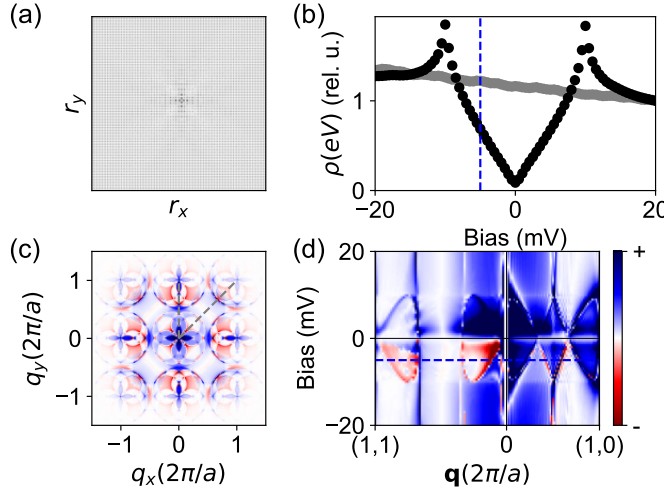

Figure 10: **QPI of a nearest-neighbour tight-binding model with a superconducting order parameter with $d_{x^2-y^2}$ symmetry.** (a) $\rho(\mathbf{r}, E)$ map with a defect in the centre. (b) spatially averaged tunneling spectrum $\langle\rho(\mathrm{r}, E)\rangle_{\mathbf{r}}$ for the superconducting state (black) and for the normal state (grey), showing the superconducting gap. (c) Phase-referenced Fourier transformation $\tilde{\rho}_{\mathrm{PR}}(\mathbf{q}, E = -5\mathrm{meV})$ calculated from eq. 19, indicated by the blue dashed line in (b). (d) Cut through the map of the density of states $\tilde{\rho}(\mathbf{q}, E)$ (a, c) along the path indicated in (c). The energy of the PR-FFT in (c) is indicated by a blue dashed line.

surface system, showing clearly the topological gap and the surface state in the bulk gap, fig. 9(c). QPI calculations show the bulk band gap Fig. 9(d), and for the surface calculation (Fig. 9(e)) no clear signature of any particular scattering vector due to the suppression of back scattering, as expected for this system.

## 4.7 Bogoliubov quasiparticle interference

One of the strengths of quasiparticle interference is that it is one of the few methods that allows phase-sensitive characterization of superconducting order parameters. This can be achieved by acquiring spectroscopic maps $g(\mathbf{r}, eV)$ in the energy range of the superconducting gap and then calculating the phase-referenced Fourier transformation (PR-FFT) [36,37],

$$\tilde{g}_{\mathrm{PR}}(\mathbf{q}, eV) = \tilde{g}(\mathbf{q}, eV) \cdot \left(\frac{\tilde{g}(\mathbf{q}, |eV|)}{|\tilde{g}(\mathbf{q}, |eV|)|}\right)^{-1}. \tag{19}$$

The resulting map, $\tilde{g}_{\mathrm{PR}}(\mathbf{q}, eV)$ (or, from a calculation, $\tilde{\rho}_{\mathrm{PR}}(\mathbf{q}, E)$), encodes whether a scattering vector connects Fermi surface sheets with the same sign or with opposite sign of the superconducting order parameter, if one assumes that the scattering is non-magnetic. In particular, for multi-orbital systems, the Bogoliubov QPI needs to be modelled to extract information about the symmetry of the order parameter from experimental data. Here, we demonstrate the basic functionality of `calcQPI` to simulate Bologjubov QPI using a nearest neighbour tight-binding model to which the superconductivity has been added, using a $d_{x^2-y^2}$ order parameter. Fig. 10(a) shows the resulting real space map at 20meV, as well as in Fig. 10(b) the corresponding tunneling spectrum in the superconducting and normal state. The PR-FFT of the map shows a rich structure governed by the symmetry of the order parameter, Fig. 10(c,d). We refer the reader for a more detailed discussion to ref. [13], where we also discuss how supercon-

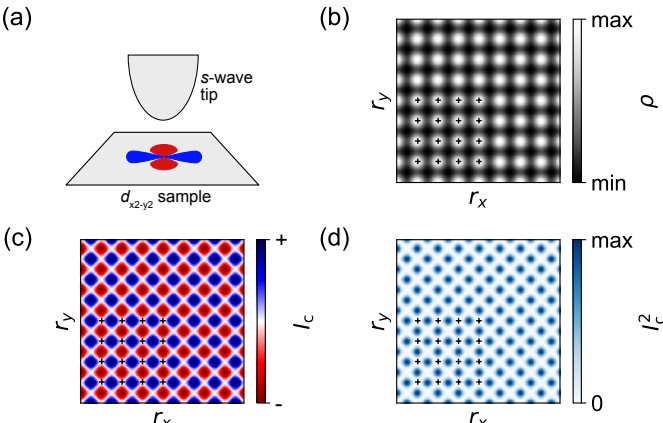

Figure 11: **Spatial modulation of supercurrent $I_c$ in Josephson STM.** (a) Sketch of experimental setup, here with an $s$-wave superconducting tip and a $d$-wave superconducting samples. (b) $\rho(\mathbf{r}, V)$ with the atomic positions indicated by small crosses, showing that the atomic contrast is highest on top of the atoms. (c) Spatial map of $I_c(\mathbf{r})$, showing the local $d$-wave lobes around the atoms. In practice, experiments do not pick up the sign but only the magnitude of $I_c$. (d) Shows a map of $I_c(\mathbf{r})^2$, where the sign-character of the order parameter is lost, but the maximum $I_c(\mathbf{r})$ would be seen on the bridge sites between atoms. The calculations are consistent with ref. [27].

ducting gaps obtained from RPA or FRG calculations can be introduced into the tight-binding model.

The example here is only meant to illustrate calculation of BQPI for a simple model, but can straightforwardly be modified to make the results more relevant for, e.g., cuprate superconductors, by setting the orbital character to $d_{x^2}$ by `orbitals=(dx2);`. In this case the results look more consistent to previous work specifically aimed at modelling STM measurements in the cuprates [5, 7].

## 4.8   Josephson STM

In recent years, Josephson STM has emerged as an tool to obtain information about spatial variations of the superconducting order parameter as well as its symmetry. For measurements on quantum materials, typically either a tip made from an $s$-wave superconductor is used [38], or a tip rendered superconducting by picking up material from a sample [39]. The technique then measures the Josephson critical current, or, more specifically, $I_c R_N$. $I_c$ can be calculated within `calcQPI`, using superconducting tight-binding models. Fig. 11(a) shows the geometry simulated here: an $s$-wave tip probing the condensate here of a $d$-wave superconductor. The differential conductance which would be obtained in a spectroscopy map is shown in Fig. 11(b). Fig. 11(c) and (d) show the Josephson critical current $I_c(\mathbf{r})$ and $I_c(\mathbf{r})^2$, respectively, showing that the maximum critical current is measured between the atoms. We note that the results are very similar to those reported previously [27].

## 4.9   Realistic calculation for a multi-band model

Finally, we present, in Fig.. 12, a calculation based on DFT-derived wave functions and a tight-binding model, including spin-orbit coupling, that has been fitted to ARPES and STM data [14]. We show here only the final result, which demonstrates the good agreement that

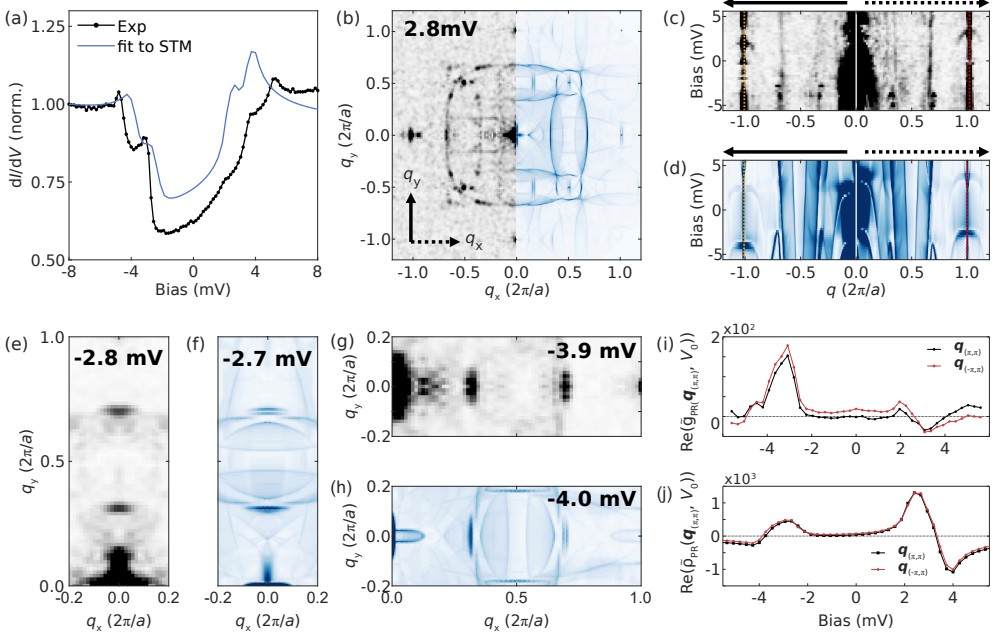

Figure 12: **Comparison of QPI calculations for a multi-band system with experiment.** (a) Comparison of experimental and calculated tunneling spectrum of the clean surface of $Sr_2RuO_4$ for a tight-binding model that has been fit to experimental data [14]. (b) Left half: quasiparticle interference image at $V = 2.8$mV, right half: calculations using model fitted to experimental data at the same voltage. (c, d) cuts through the atomic peaks along $q_y$ (left) and $q_x$ (right) from experiment (c) and calculations for the tight-binding model fitted to the QPI (d). The solid and dotted arrows indicate the $q_y$ and $q_x$ directions as in (b). (e, f) Detail of the QPI along the vertical and (g, h) horizontal direction, revealing the excellent agreement. The maxima along the x- and y directions occur at slightly different energies due to the nematicity. (i, j) Comparison of real part of the PR-FFT, $\tilde{g}_{PR}(\mathbf{q}_{(\pi,\pi)}, eV_0 = -3\text{meV})$, at the checkerboard peaks as a function of energy for the experimental data (i) and the calculations (j).

can be achieved, as well as an example for a multi-band system, which contains bands of both spin characters. The tight-binding model considered here is projected from a DFT calculation of a free-standing layer of $Sr_2RuO_4$, obtained by projection onto the $t_{2g}$ manifold of the Ru $4d$ shell and using a 2-atom unit cell because of the in-plane $RuO_6$ octahedral rotations. To match the ARPES data around the VHs near the $\overline{M}$ point, the band structure is renormalized by a factor of 4 and 175meV of spin-orbit coupling is added. Matching it to the QPI data requires a small nematic term and a further small shift, for details we refer the reader to ref. [14]. Fig. 12(a) shows the final result of the averaged real-space cLDOS as a function of energy in comparison with the experimental tunneling spectrum, showing excellent agreement. Panels (b-d) show detailed comparison of the QPI, again with a very good agreement between the two. The calculations reproduce the VHs around the atomic peak that is split by nematicity, as well as the intensity distribution along the high symmetry directions, see fig. 12(e-h). Notably, the calculation reproduces a checkerboard pattern that is also experimentally observed (Fig. 12(i,j)), reproducing not only the overall energy dependence of the intensity but also of the phase of these peaks.

## 5   Conclusion

quasiparticle interference is a powerful tool to aid in the understanding of the electronic structure of quantum materials, providing valuable complementary information to ARPES, such as information about the unoccupied electronic states, and in particular in parameter regimes that are not accessible by ARPES, e.g., in magnetic fields or at ultra-low temperatures. However, because QPI probes a correlation function, the interpretation is not straightforward and often requires advanced modelling. `calcQPI` attempts to fullfil this role, by providing a means for realistic simulation of quasiparticle interference, incorporating the role of the tunnling matrix element effects. We have demonstrated in a series of papers the high quality of agreement with experiment one can obtain from these calculations [9, 12, 14] as well as their usefulness in understanding how QPI is impacted by subtle changes in the electronic structure [11]. We note that while the examples shown here are all for tetragonal lattices, the code has successfully be used to study QPI on hexagonal lattices, for example for graphene [15]. While not shown here, `calcQPI` also allows for the calculation of the unfolded spectral function [15], which enables comparison to data from both STM and ARPES, and thus paves the way to simultaneous fitting of data from multiple techniques to refine the parameters of a tightbinding model. This currently needs to be done manually, but as a next step multi-dimensional fitting, e.g., through Bayesian fitting would be desirable and come into reach. As a future improvement of the fidelity of the simulated quasiparticle interference, the inclusion of a realistic self-energy of the system, for example from Dynamical Mean Field Theory [40, 41], is desirable, which should then properly describe the decrease of the quasiparticle lifetime resulting in broadening of the QPI signal with increased bias voltage, as is typically seen in quantum materials.

## Acknowledgements

We gratefully acknowledge all those who have contributed to testing and improving the code, including Olivia Armitage, Izidor Benedičič, Siri Berge, Rebecca Bisset, Uladzislau Mikhailau, Bruno Sakai, Rashed AlHamli, Elle Debienne, Hugo Decitre, Dylan Houston, Seoyhun Kong, Victoire Morriseau, Alexandra Munro and Weronika Osmolska. We further acknowledge insightful discussions with Andreas Kreisel and Peter Hirschfeld. PW acknowledges many stimulating discussions about quasiparticle interference with Shun Chi.

We acknowledge funding from the Engineering and Physical Sciences Research Council (EP/S005005/1, EP/R031924/1) and the Leverhulme Trust through RPG-2022-315. LCR acknowledges support through the Royal Commission for the Exhibition of 1851 and CDAM from the Federal Commission for Scholarships for Foreign Students for the Swiss Government Excellence Scholarship (ESKAS No. 2023.0017) for the academic year 2023-24. The work benefited from computational resources of the Cirrus UK National Tier-2 HPC Service at EPCC (http://www.cirrus.ac.uk) funded by the University of Edinburgh and EPSRC (EP/P020267/1), of Archer2 (https://www.archer2.ac.uk), the High-Performance Computing clusters Kennedy and Hypatia of the University of St Andrews and Marvin of the University of Bonn.

| Keyword | Parameter |
|---------|-----------|
| Input files | |
| tbfile | file name of file containing tight-binding (TB) model in Wannier90 format. |
| Output files | |
| qpifile | name of main output file, e.g. for QPI, in .idl format. |
| logfile | name of file to save output log (default: console). If a file is specified, all output is sent to that file, including any error messages. |
| wffile | name of file to save output wave functions, in .idl format (see app. C). This is useful to check whether the wave functions specified as Slater orbitals are correctly parsed. |
| Controlling output | |
| output | output mode for QPI (default: wannier). Available options: wannier, josephson, spf, uspf, nomode. |
| Dimensions of output array for QPI | |
| lattice | total number of unit cells that determines the lateral size of the area which is calculated (default: 201). |
| oversamp | number of points per unit cell (i.e. oversampling) (default: 4), this should be sufficiently large so that the Nyquist theorem is fulfilled for the atomic resolution in the image, so will depend on the number of atoms in the unit cell and their positions. If a wave function file is used, the value must match the one used to generate the wave function file. |
| energies | energy range of the simulated map in eV, inclusive of the limits (default: -0.1, 0.1). |
| layers | number of energy values in the energy range specified by energies at which the map is calculated (default: 21). |

Table 1: List of keywords recognized by calcQPI to specify the input and output files, and the dimensions of the output array.

## A    Command reference for calcQPI

calcQPI is run by providing the name of an input file on the command line, i.e. with calcqpi filename. The file filename contains a sequence of keywords. The order does not matter. The general syntax is

```
keyword=value;
```

Strings which refer to names should be enclosed in quotation marks ". Some commands require no input, or numbers, strings, vectors or lists as parameters, which should be written as follows.

```
command=;
number=1.0;
string="filename";
vector=(1.0,2.0);
list=("file1","file2");
```

In table 1 is a list of keywords recognized by calcQPI for the input and output files is provided.

| Keyword | Parameter |
|---|---|
| **Parameters for Green's function** | |
| green | mode for calculating the Green's function (default: `normal`). Available options: `normal`, `surface`, `bulk`. Calculations with the `normal` Green's function are significantly faster than those using the `surface` or `bulk` Green's function. For `surface` and `bulk` Green's functions, it is usually inefficient to do the calculation on a GPU. |
| stbfile | name of file containing TB model of surface layer for a `surface` Green's function calculation. If none is specified, the TB model specified in `tbfile` is used, so that the same TB model as for the bulk is used in the calculation. |
| epserr | convergence criterion for the iterative calculation of the surface Green's function. The iterative loop ends when both $\|\alpha_N\|^2$ and $\|\beta_N\|^2$ are smaller than the specified value (default: 1e-5, see also section 2.2.2). For larger matrices (systems with more bands), it is advisable to use a larger value to achieve similar precision as for small matrices. |
| eta | broadening parameter $\eta$ in eV (default: 0.005) which enters in the Green's function, eq. 3. |
| kpoints | lateral size of the $k$-point grid, if not specified this will be set to `lattice`. For QPI calculations, one should chose a value such that the energy difference between the states within a band at different $k$ values is smaller than the broadening parameter $\eta$, so for bandwidth $D$, typically `kpoints`$> \frac{D}{\eta}$. |
| spin | if `true` assumes that half the bands are spin up, and the other half spin down (default: `false`). |
| **Parameters of *T*-matrix calculation** | |
| fermi | Fermi energy in eV (default: 0.0). This allows for a chemical potential shift to be applied to the tight-binding model before any calculation is carried out. |
| scattering | specify the diagonal of the scattering matrix $\hat{V}_\sigma$ as list of factors. This is typically 1 or 0, but also other values can be chosen to set specific scattering potentials for specific orbitals (default: all 1). For systems with more than one atom per unit cell, this allows to select the orbitals of a specific atom to calculate the case of a point-like scatterer. |
| phase | scattering phase $V_0$, as a complex number (default: (1,0)). |
| **Parameters for the continuum LDOS** | |
| window | window of size n, defining the real space cut off. (default: 2). If `window` is too small (i.e. the value at the boundary of the `window` is more than 10% than the maximum value), a warning will be issued. If `window` is set to zero, the value will be estimated automatically (only for Slater-type orbitals specified in the configuration file). If a wave function file is specified, this value must match the one used to generate the wave function file. |
| threshold | threshold value, default: NAN. If a value is specified, a list of the wave function products where the value is larger than the threshold is used to calculate the continuum transformation (see section 2.4.1). |

Table 2: List of keywords recognized by `calcQPI` to define what Green's function to use, and the parameters for the continuum transformation.

| Keyword | Parameter |
|---------|-----------|
| Parameters for wave function overlaps loaded from a file | |
| orbitalfiles | list of input files used for Wannier functions, the files need to be in the .xsf format written, e.g., by wannier90. One file needs to be provided for each orbital. |
| idlorbitalfile | .idl file containing the Wannier functions (typically generated by mkwavefunctions or using the mkwffile). This file contains a cut at constant height through the wave function generated, e.g., from DFT calculations or using Slater orbitals. The file needs to contain one layer per orbital of the model, and must have been generated with the same settings of window and oversamp as the calculation. |
| Parameters for wave function overlaps using Slater orbitals | |
| radius | radius that defines the size of the orbital functions. It is in units of the lateral size of the unit cell, where 1 means an orbital with radius equal to the unit size (default: 0.5). |
| angle | rotate all Wannier functions by specified angle (default: 0.0). This is for cases where the axis of the coordinate system in which the orbitals are defines is not the same as the axis of the unit cell, as is, e.g., often the case for unit cells with more than one atom (e.g. often angle=45 for a two-atom unit cell square lattice). |
| anglearr | list of additional angles by which individual Wannier functions are being rotated (default: none). This is a list of values that will be added to angle. |
| prearr | array of prefactors that will be multiplied to each Wannier function (default: all 1) |
| orbitals | list of orbitals used for Wannier functions. calcQPI has Slater-type orbitals predefined. Supported orbitals are s, px, py, pz, dxy, dx2, dr2, dxz, dyz, fy3x2, fxyz, fyz2, fz3, fxz2, fzx2, fxx2. If the number of wave functions is half the number of bands of the tight-binding model, the orbitals are simply duplicated assuming that the calculation is spin-polarized, otherwise if not enough orbitals are specified, the unspecified ones are set to zero and hence ignored in the continuum transformation. |
| pos[0], ... | positions of the orbitals given as (x,y,z) coordinates, normalized to the size of the unit cell (default: (0.0, 0.0, 0.0)). The coordinates need to be specified for each orbital in the TB model (including spin up and spin down). If the z-coordinate is omitted, it is set to zero. |
| zheight | tip height above top-most atom (when using Wannier functions from DFT, negative height is below bottom atom). Also used for atomic-like orbitals as height above zero, the center of the orbitals. |

Table 3: List of keywords recognized by calcQPI for loading or creating wave function files.

| Keyword | Parameter |
|---|---|
| Parameters for bandstructure histogram | |
| bsfile | output file for band structure |
| bslattice | size in pixels in the first Brillouin zone (default: 201) |
| bsoversamp | n-times oversampling (default: 8) |
| bsenergies | energy range in eV (default: -0.1, 0.1) |
| bslayers | number of energy layers (default: 21) |
| Parameters for DOS output | |
| dosfile | filename for output of total density of states |
| dosenergies | energy range in eV (default: -0.1, 0.1) |
| doslayers | number of energy layers (default: 101) |

Table 4: List of keywords recognized by `calcQPI` for the calculation of density of states and band structure histograms.

| Keyword | Parameter |
|---|---|
| Parameters for Bogoljubov QPI | |
| scmodel | if = `true`, assumes the model is superconducting and only the first half of the bands in the TB model is considered for the calculation of the density of states. This also ensures the negative sign for $k$ for the anti-particle sector of the Nambu Hamiltonian. |
| magscat | fraction of magnetic scattering. Options: $= 0$, only non-magnetic scattering; $= 1$, only magnetic scattering (default: 0). |
| Parameters for Josephson mode | |
| deltat | gap size of the tip in eV (default: 0.001). |
| etat | energy broadening for the tip in eV. The default is the same as sample, specified by `eta`. |

Table 5: List of keywords recognized by `calcQPI` for superconducting models.

## B   Command reference for `mkwavefunctions`

`mkwavefunctions` is an ancillary tool for creating the wave function overlaps in a variety of ways. It offers more parameters than `calcQPI` itself, in particular for example for non-square lattice geometries. It also allows to manipulate wave functions, for example enabling the use of DFT-derived wave functions for larger unit cells.

## C   Output file format

The standard file format used by `calcQPI` is a binary format that is referred to as '.idl'-file. The file contains a text header of 12 lines which contain the meta data, and then the binary data as a 3D array of single precision floats. The 12 lines in the beginning are:

```
Comment
Mon Apr 12 23:45:38 2021
320
320
40
2
2
-1
-1
-0.4
0.4
0
[binary data from here]
```

A detailed account of each line is provided in table 7.

## D   Obtaining wave functions from DFT calculations

This section provides a brief description of how to obtain wave functions from DFT calculations, based here on VASP, however similar steps are required when using Quantum Espresso. As an example, here the case of $Sr_2RuO_4$ is discussed, which works particularly well because of its quasi-two-dimensional electronic structure. One starts from a crystal structure that is representative of the surface layer and performs a slab calculation. In the case of $Sr_2RuO_4$, this requires a two-atom unit cell and converges well with a $k$-point set of $7 \times 7 \times 1$ $k$-points. After converging the charge density from a self-consistent field calculation (and possibly after relaxing the crystal structure), one plots the band structure from a non-self-consistent calculation along a high-symmetry path as reference, in the case of $Sr_2RuO_4$ along the path $\Gamma - M - X - \Gamma$. The next step is to project the DFT model onto a tight-binding model using Wannier90. For this, one needs to define how many bands are going to be projected, which projections should be done, and the $k$-points. The parameters of the projection need to be optimized (typically the disentanglement window (`dis_win_min`, `_max`) and the frozen window (`dis_froz_min`, `_max`)), and one should check that the wannier90 projection converges. Often, good results are obtained using projected wave functions (using `num_iter=0`), rather than maximally localized wave functions. Once a good representation of the DFT bandstructure is achieved, which is checked by plotting the band structure in `wannier90_hr.dat`, one can generate the wave functions using `wannier_plot=.true.` in the `wannier.win` file (this should be

| Keyword | Parameter |
|---|---|
| **Input and output files** | |
| `logfile` | output log to logfile (default: cout) |
| `tbfile` | name of tightbinding model |
| `mkwffile` | output wave functions to the specified file |
| **Parameters for QPI calculation** | |
| `oversamp` | n-times oversampling (default: 4) |
| `window` | window of size n (default: 0 (auto), if too small, a warning will be issued) if no window specified, the value will be estimated automatically |
| **Parameters for Slater-type orbitals** | |
| `radius` | use radius for size of orbital functions (default: 0.5) |
| `radarr` | an array of radii, one for each orbital |
| `angle` | rotate wannier functions by specified angle (default: 0.0) |
| `anglearr` | additional angle by which individual Wannier functions are being rotated (default: none) |
| `phiarr` | out-of-plane angle by which individual Wannier functions are being rotated (default: none) |
| `prearr` | array of prefactors for orbitals (default: all 1) |
| `orbitals` | list of orbitals used for Wannier functions |
| `pos[0], ...` | positions of the orbitals (default: 0,0) |
| `zheight` | height of z-layer above top-most atom (when using Wannier functions from DFT, negative height is below bottom atom; used also for atomic-like orbitals) |
| `basisvector[0], [1]` | basis vectors for a basis where they are not just (1,0) and (0,1) |
| **Input wave functions from DFT or other sources** | |
| `orbitalfiles` | list of files used for Wannier functions |
| `idlorbitalfile` | idl file containing the Wannier functions |
| **Input wave functions from DFT for further modification** | |
| `modorbitalfiles` | list of files for Wannier functions |
| `modpos[0],...` | shift position of input wave function |
| **Interpolation of orbitals** | |
| `idlorbitalfiles` | list of .idl `"filename1.idl"`, `"filename2.idl"`, ... files containing the Wannier functions for interpolation. |
| `values=(x1,x2,...)` | values corresponding to `filename1`, `filename2`, ...,. If no values are provided, the $x1...xn$ are mapped uniformely on the interval $0...1$ |
| `interpolate` | value at which the model is to be interpolated |
| `interpmethod` | interpolation method. Available options: `linear`, `polynomial`, `akima`, `cspline`, `cspline_periodic`, `akima_periodic`, `steffen`. |
| **Reorganizing wave functions** | |
| `double` | double orbitals for spin-polarized calculation |
| `reorder` | reorder - provide list of numbers which contains indices to the original order. |

Table 6: List of keywords recognized by `mkwavefunctions` to create the wavefunction file.

| Value | Parameter |
|---|---|
| Comment | Comment. |
| Mon Apr 12 23:45:38 2021 | date when file was created. |
| 320 | number of pixels in x-direction. |
| 320 | number of pixels in y-direction. |
| 40 | number of energy layers. |
| 2 | lateral physical size in x. |
| 2 | lateral physical size in y. |
| -1 | offset in x. |
| -1 | offset in y. |
| -0.4 | low limit of energy range in eV. |
| 0.4 | high limit of energy range in eV. |
| 0.0 | setpoint current. |

Table 7: Line-by-line description of the output file format.

done using the patched version of Wannnier90 [19]). The results are the wave function files `wannier90_#####.xsf`.

Finally, it is convenient to convert the wave functions from Wannier90 into the 'idl' format using `mkwavefunctions` to feed into a QPI calculation in `calcQPI`. We provide input files for VASP, Wannier90 and `mkwavefunctions` to project a model for a single monolayer of $Sr_2RuO_4$, including octahedral rotations as they are found in the surface layer, with the examples.

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
