# Peer review of "calcQPI: A versatile tool to simulate quasiparticle interference"

_SciPost Physics Codebases, doi:SciPost Phys. Codebases 61 (2025) , SciPost Phys. Codebases 61-r1.0 (2025)_

## Round 2 · Referee Report · Andreas Kreisel (Referee 1) · 2025-10-4

Report
The authors have incorporated changes to the manuscript according to the comments of both referees such that minor mistakes, typos or imprecise statements are removed. The addition of the procedure for obtaining ab-initio Wannier functions is certainly valuable and rounds up the presentation of the calcQPI code.
Looking into the changes in detail, I found a few small issues that could be fixed:
1) Figure 8 has been updated (single impurity vs. double impurity calculation). In the caption it is written “(b) Bulk (left half) and surface (right half) spectral function of the model”. Seems that in the new figure, bulk and surface parts are overlayed and plotted in “red” and “black” shading. Minimally, the caption has to be changed; better would be actually the splitting of the figure in two halves with plotting bulk and surface spectral function as it was in the previous version.
2) Figure 2; the equation in the box seems still not to be identical to Eq. (7). G_0 does only have one spatial index and the second Green function should have argument “ -R’ ”.
Recommendation
Publish (surpasses expectations and criteria for this Journal; among top 10%)

---

## Round 2 · List of Changes

• We have updated the model and figure for the Rashba-spin-orbit split system, because due to a typo the Hamiltonian in the examples was non-hermitian. The results look identical.
• We have fixed minor bugs in the code which came up while the review process was going on. These only affect the interface, but none of the core components.

---

## Editorial Decision

published